# Cross-Method-Based Analysis and Classification of Malicious Behavior by API Calls Extraction

**Bruce Ndibanje [1]** , **Ki Hwan Kim [2]**, **Young Jin Kang [2]**, **Hyun Ho Kim [2]**, **Tae Yong Kim [3]** and **Hoon Jae Lee [3],\***

[1] Research and Development Center, Cyber Threat Intelligence Lab, YangJae Innovation Hub, 114 Taebong-Ro, Seocho-Gu, Seoul 06764-601, Korea; ndibabruce@gmail.com

[2] Department of Ubiquitous IT, Graduate School of Dongseo University, Sasang-Gu, Busan 617-716, Korea; dgend90@gmail.com (K.H.K.); rkddudwls55@gmail.com (Y.J.K.); feei_@naver.com (H.H.K.)

[3] Division of Computer and Engineering, Dongseo University, Sasang-Gu, Busan 617-716, Korea; tykimw2k@gdsu.dongseo.ac.kr

\* Correspondence: hjlee@gdsu.dongseo.ac.kr

**Abstract:** Data-driven public security networking and computer systems are always under threat from malicious codes known as malware; therefore, a large amount of research and development is taking place to find effective countermeasures. These countermeasures are mainly based on dynamic and statistical analysis. Because of the obfuscation techniques used by the malware authors, security researchers and the anti-virus industry are facing a colossal issue regarding the extraction of hidden payloads within packed executable extraction. Based on this understanding, we first propose a method to de-obfuscate and unpack the malware samples. Additional, cross-method-based big data analysis to dynamically and statistically extract features from malware has been proposed. The Application Programming Interface (API) call sequences that reflect the malware behavior of its code have been used to detect behavior such as network traffic, modifying a file, writing to stderr or stdout, modifying a registry value, creating a process. Furthermore, we include a similarity analysis and machine learning algorithms to profile and classify malware behaviors. The experimental results of the proposed method show that malware detection accuracy is very useful to discover potential threats and can help the decision-maker to deploy appropriate countermeasures.

**Keywords:** malware classification; behavior analysis; machine learning; feature selection; API; static analysis; dynamic analysis

## 1. Introduction

Cybersecurity threats are growing and rapidly adapt to new opportunities in cyberspace. The inter-connectivity of devices and services via high internet speeds make it easy for cybercriminals to operate remotely from overseas and remain unidentified online. In this context, it is, therefore, challenging to identify and trace the malware origin of such crime. Moreover, with the social networks rapid development, the spread of malware by hackers is more frequent than before as they widely use various kinds of social networks [1].

By its definition, malware [2], also known as "malicious software", is a software created by an attacker to compromise the security of a system or privacy of a victim. Malware includes computer viruses, worms, Trojan horses, spyware. Additionally, the quantity and types of malware [3] have increased, and they are challenging cybersecurity experts, law, and forensics examiners [4–7]. To launch and spread these attacks, some technologies facilitate them and can avoid security systems. The Onion Router (the ability to apply multiple proxy routing that inhibits the ability to trace-back), Obfuscation (a

term used to describe the modification of a program to disguise its purpose), Dynamic Domain Name System (DDNS or DynDNS), and Virtual Private Network (VPN) services are all capable of facilitating cybercrime. These methods employed by the cybercriminals, enable them to deepen their criminal activities without being easily intercepted by network security and traffic analysis. Furthermore, they take advantage of advanced technology to develop new malware or variants that give them more power to effectively achieve the propagation of malware and stay anonymous online as long as possible. In the case of attacks or a successfully committed cybercrime, it is laborious to investigate such malicious activities. When performing an investigation on cybercrime [8], it is mandatory to raise "5 Ws and 1H" basic questions. When, Where, What, What, Who, and Why should be asked during the investigation. Where and when was the crime committed? What technologies were used? Who was behind the crime and why was the crime committed? How was the crime committed?

Signature-based detections such an antivirus (e.g., Avast, Symantec, etc.) and anomaly-based are the two main techniques used for malware detection systems. For the first method, experts in InfoSec have daily tasks to generate signatures to be used by anti-virus engines to detect known malware. Nevertheless, the signature-based detection is not able to detect unknown threats and, thereafter, is not effective against "zero-day attack" [9]. Additionally, there is another security challenge caused by packed or obfuscated malware [10] making it difficult for digital forensic examiners to express the true purpose of malware, thus affecting the detection accuracy. For this reason, the anti-virus method suffers from two hindrances; first, high false positive and second, high false negative. Technically, it identifies benign files as malware in the former case and it fails to detect malware in the latter case [11,12].

Therefore, in our research, we focus on both methods and thus, we can overcome the problem of unknown malware detection. In this case, features extraction such as Application Programming Interface (API) call sequences and code obfuscation features will be used to study the behavior of the sample execution and malware such as file operations creation and deletion of process. This method is based on dynamic analysis to classify the malware variants with a similar behavior context. The API sequence methods give good feature selection to use to compute the similitude or similarity of two malware variants because they represent the behavioral features of malware. In this paper, we opt to use an API call sequence to compute the similarity and classify the files into malware and benign.

The novelty of this paper is based on the use of a machine learning algorithm while computing the similarity to classify the malware in addition to an algorithm of API frequency and sequence for making a dataset made of feature selections. At first, we use the static method, which consists of software of malicious code analysis without execution. To overcome obfuscated malware, we have used IDA Pro and PEid. Second, we use the dynamic analysis where we set up a virtual environment to run the samples. In the course of the experiment, we implemented a process to remove redundant sequences and expand on the design of the feature selection database. Finally, we calculate the similarity for classifying the malware using an API sequence and the Microsoft Developer Network (MSDN). The experimental results of the proposed method show that malware detection accuracy is very useful to discover potential threats and can help the decision-maker to deploy appropriate countermeasures. The dataset used in our experiment is given in Table 1 while Table 2 gives the summary about the malware categories and the results are reported in Tables 3–6.

The rest of the paper is organized as follows: Section 2 presents the research literature on malware analysis and detection methods. Our method and implementation are detailed in Section 3. The experimental procedure and results are reported in Section 4 before concluding our work in Section 5.

## 2. Related Work

Extensive research has been proposed and new research is still performed regarding the provision of solutions for malware detection systems [13,14]. For instance, in the case of known malware, content signatures-based methods that map samples of activities against known malware have been proposed [15,16]. Nevertheless, these methods have weaknesses when presented with obfuscated

malware, metamorphic or polymorphic techniques to hide malware, and unknown malware. However, different solutions have been proposed to overcome such techniques [17–20].

Moreover, dynamic analysis [21] was proposed to observe dynamic behaviors and features for packed malware. In this context, a virtual environment is used to monitor the real-time execution of malicious code and dynamically analyze the behavior of extracted features such as system calls, memory usage, and network behavior. The detection of unknown malware that shows similar behavior is also possible using this method [22]. The API call analysis and control flow are the two major dynamic analysis methods [23,24]. Both static and dynamic methods can be used to extract API calls information that reveals how malware behaves. From the executable malware, the Portable Executable (PE) format is the essential element used to extract the API list using a static approach [25–28]. By running the executable files in a sandbox, the called API can be observed with dynamic methodology [29–33]. The main difference between existing works and ours, is that our approach combines both static and dynamic analysis methods in order to increase the detection rate and overcome static analysis weakness. Furthermore, our approach includes a similarity analysis to classify the files between benign and malware files.

## 3. Proposed Method and Implementation

In this section, we describe the method used for malware classification based on API call lists made from dynamic and static analysis, as shown in Figure 1, where the dash arrow line gives the direction to the data as input into Cuckoo S-Box and the thin and thick arrows lines show different directions from a given component to another. The proposed framework mainly consists of three modules: a database of malware and the PE file, modules of static and dynamic analysis, and a module of classification by similarity analysis. The module concerns all datasets used as input to the static and dynamic modules. The static analysis consists of the de-obfuscation of packed malware in order to know the packers names and the contribution level of each of them. For this, a simple python code have been used and one of the codes (to identify malware samples which are related to one another) used in this research is given in Appendix B. In addition, IDA Pro and PeStudio are used to enhance the de-obfuscation steps. These tools are mainly used in the reverse engineering of malware. They help us to identify logic and arithmetic and address manipulation and flow control features from malware as opcodes. A dynamic analysis module consists of a virtual environment set up by Cuckoo Sand-Box to run the executable files of malware without infecting the rest of the system. The API call list database is obtained from the network behavior and other important features. The final module concerns the malware classification process using Machine Learning algorithms. In our research, we have used a similarity analysis based on API sequence, which includes benign and malware files. The comprehensive API reference, which is the Microsoft Developer Network (MSDN), is used to operate a match and classify files between malware and benign.

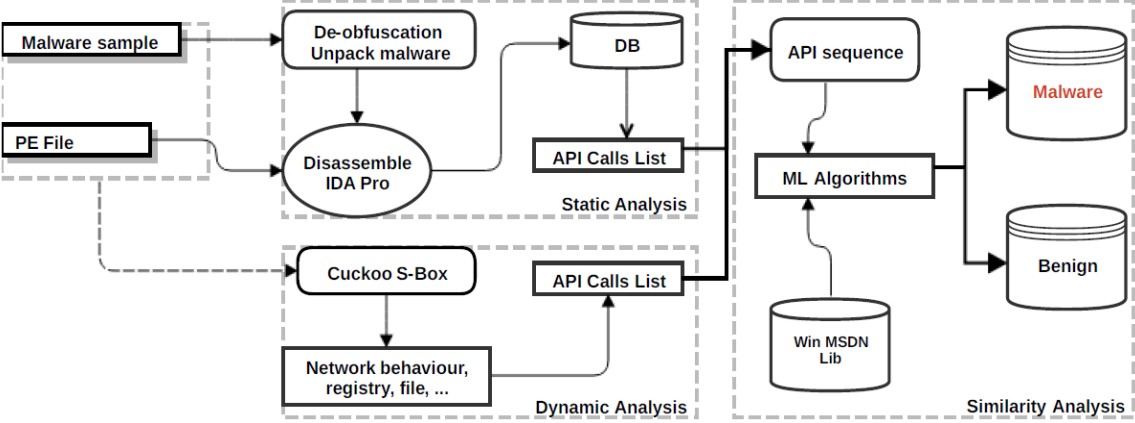

**Figure 1.** Framework for the analysis and classification of malware. API: definition.

### 3.1. Dataset and Preprocess

The dataset used in our experiment was built using a VirusShare source [34] with an approximate number of 78,568 executables files. Among them, 61,354 are different malware (78% of the total number). The rest of the data were are identified as benign (12% of the number) from different sources. Some of the malware samples were obfuscated while others were not or unknown. The preprocessing step is to overcome the packed or obfuscated malware. Solutions from known tools such as IDA Pro (By Hex-Rays sa, Liege, Belgium), PEiD (PEiD is a small application which is used to detect common packers, cryptors and compilers), and PeStudio (PEStudio is a network protocol analysis and security auditing tool for Windows which allows you to apply scripts to winsock calls it is used by Computer Emergency Response Teams (CERT) and Labs worldwide in order to perform Malware Initial Assessment.) have been used to unpack the malware samples. The advantage of de-obfuscation is that the statistical analysis is performed accurately. A summary of the dataset is given in Table 1.

**Table 1.** Summary of dataset created as per VirusTotal.

| Type | Source | Quantity | Percentage |
|------|--------|----------|------------|
| **Malware** | VirusShare | 61,354 | 78% |
| **Benign** | Various locations and made from our lab | 17,214 | 12% |

### 3.2. Static Analysis Process

The static analysis consists of software for malicious code analysis without execution in order to determine the pattern attack of the malicious program and its capabilities to harm the system. This process helps us to identify the byte-sequence n-grams, a syntactic library call, control flow graph, and all types of non-executable and executable files. However, the static analysis has limitations due to packed or obfuscated malware. Similar methods have been proposed such as Dynamically Generated and Obfuscated Malicious Code (DOME) which is a host-based technique for detecting several general classes of malicious code in software executables [35]; however, it was unable to handle packed malware.

In this research, to overcome the challenge of packed malware, we have used the dissembler tools such as IDA Pro (IDA is the **I**nteractive **D**is**A**ssembler: the world's smartest and most feature-full disassembler, which many software security specialists are familiar with) [36] and PeStudio [37]. Figure 2 shows the distribution of the packers names from our dataset.

The experiment has been conducted over 61,354 malware files, and we used the IDA Pro to translate a program into its equivalent high-level-language program given the content of the binary and PEid [38], a specialized software to unpack the malware. As given in Figure 2, the result is that 43% of these malware programs are unpacked identified as "nothing", with the other identifiers accounting for approximately 57% of the total percent. From the result, we can understand that the malicious code has the ability to evade the detection system by changing their byte sequence with the aid of obfuscation techniques. The SQLite (SQLite is a C-language library that implements a small, fast, self-contained, high-reliability, full-featured, SQL database engine) [39] has been used to construct our API call lists after dissembling the malware. It is a library used by IDA Pro to allow security analysts to create plug-ins to be performed by IDA Pro.

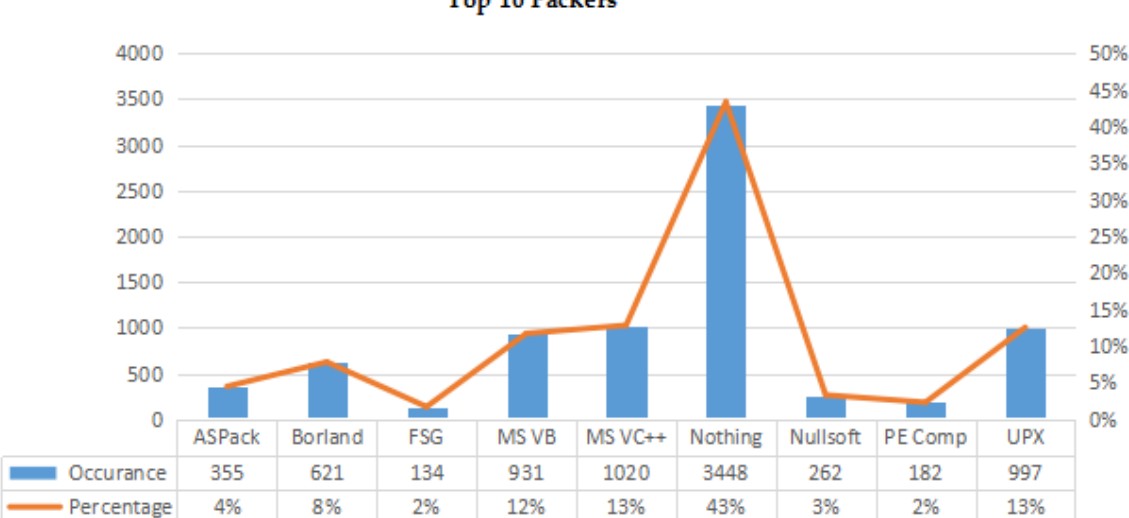

**Figure 2.** Top 10 packers from 8637 packers.

In addition, it implements an independent transactional SQL DB engine in order to generate the database (.db). During this process, by using the SQLite plugin (SQLite is a C-language library that implements a small, fast, self-contained, high-reliability, full-featured, SQL database engine) we have generated 16 tables; however, we list only eight tables as follows: Expression, modules, instructions, metainformation, callgraph, function, segments, and basic blocks. Different information from the tables regarding the binary content has been observed. The Stacks table contains the function address, the stack name, and the start and end addresses. The Function table contains all the recognizable API system calls, non-recognizable function names, and the length (start and the end location of each function). The Segments table contains information that describes each segment in an executable file, segment name (Code, Data, BSS, idata, tls, rdata, reloc, andrsrc), and the segment length. The operation code (OP) and their addresses and block addresses are contained in the instructions table. The API calls have been extracted using the function table. To operate a match and identify the windows API, we have used the reference from the Microsoft Developer Network (MSDN) which is a comprehensive API reference for working with Microsoft tools, services, and technologies. Figure 3 is a snapshot of IDA Pro in action.

```
loc_4017D4:
mov     ecx, [esp+54h+hObject]
mov     esi, ds:CloseHandle
push    ecx             ; hObject
call    esi ; CloseHandle
mov     edx, [esp+54h+var_4]
push    edx             ; hObject
call    esi ; CloseHandle
push    0               ; bFailIfExists
push    offset NewFileName ; "C:\\windows\\system32\\kerne132.dll"
push    offset ExistingFileName ; "Lab01-01.dll"
call    ds:CopyFileA
test    eax, eax
push    0               ; int
jnz     short loc_401806
```

**Figure 3.** Interactive DisAssembler (IDA Pro) dissembling an executable file.

### 3.3. Dynamic Analysis Process

This method is mainly based on the analysis of the behavior of the malware and it is different to the static method in the fact that a simulated environment is set up (virtual machine, sandbox,

simulator, etc.) to run the malware in order to extract different malware behaviors. Dynamic analysis is more effective compared to static analysis and does not require the executable file to be disassembled. It discloses the malware programs' natural behavior that is more resilient to static analysis.

A dynamic analysis method to detect malicious software has been developed by Shabtai et al. [40]. Their system detects malware by observing the network patterns of applications and the same research claims that there is pattern similarity of network traffic patterns of different applications with the same functions. A web-based android malware detection and classification system has been proposed [41] where they developed an auto-trigger view identification in addition to a droidbox structure. Android malicious software detection is outside the scope of this paper. In this research, Cuckoo SandBox has been used as it provides information on the file system, registry keys, and network traffic monitoring in a controlled environment and produces a well-formed report from Jason format. For the API call list from the malware analysis behavior, the call sequence has been considered using e-behaviors, i.e., file behavior, registry behavior, process behavior, and network behaviors. The Windows application programming interface (API) from the Microsoft Developer Network (MSDN) has been used to map and operate a match.

### 3.4. Similarity Analysis Process

Similarity analysis has been performed for malware classification carried out by implementing a distance measure in python. Different measuring techniques are used in this research such as Minkowski Distance, Cosine Similarity, Containment Broder, Canberra distance, and Longest Common Subsequence (LCS). More details are described in Section 4.3 to estimate the maliciousness of a code. However, in [42], the authors raise a problem of a "malware variant" such as Win32.Evol which has multiple variants from the same sample due to packer methods used by malware coders. However, to overcome this difficulty, different approaches based on similitude detection are available to verify if the variant is the child or grandchild of the sample. In this paper, we have used a five-methods-based distance that possessed a vector function for analyzing the similarity between two-paired vectors. More details are provided in Section 4.

## 4. Experiments and Results

This section describes the experimental procedure and results of our research. We set up a virtual environment to run and monitor any suspicious behavior from the executable programs. The malware dataset was prepared using VirusShare source with a total of 78,568 executables files. Among them, 61,354 are different malware programs. The rest of the dataset were identified as benign and from different sources. Some of the malware programs were obfuscated.

### 4.1. Environment Set Up

In order to collect information and extract the most important API functions, a virtual environment has been established. In our experiments, we have used Ubuntu 18.04 on the host machine where the Cuckoo Sandbox is also installed in addition to python scripts we have developed for the project. We set up two virtual machines using VirtualBox and installed Windows XP to analyze the malware. The IDA Pro and PEid have been installed on a 3rd virtual machine under Windows 7 and additional tools have been installed on virtual machines for malware analysis. Cuckoo submits files for analysis to Windows XP virtual machines, collects activity traces of a file, and generates different reports. Reports consist of, a dump of network traffic and csv reports which includes API functions used by file. Figure 4 gives an overview of our experiment set up where three virtual machines have been installed on Ubuntu as the host OS.

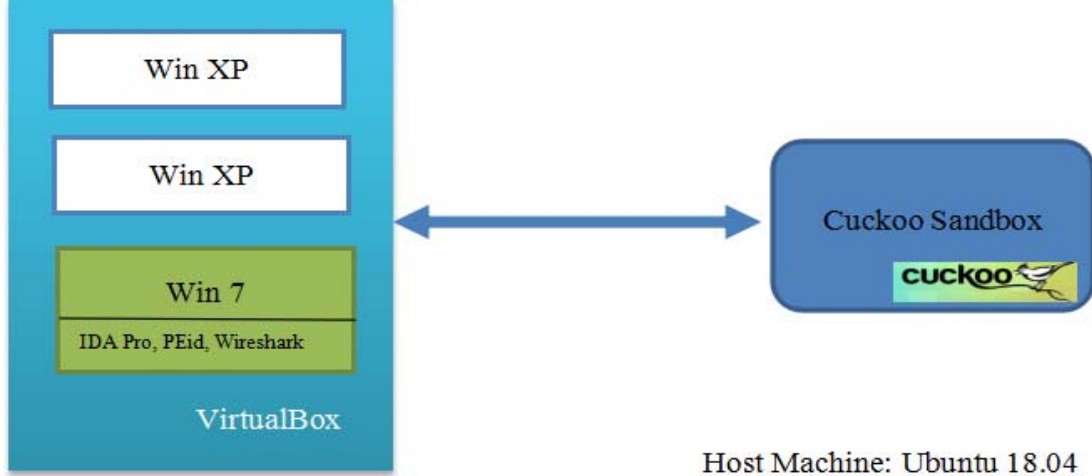

**Figure 4.** The main component of the experimental environment.

Malware detection was performed at runtime using the cuckoo sandbox [43] tool because it provides a safe environment to execute malware without infecting the whole system. A laptop with an Intel Core i7-6700HQ processor (it is a quad-core processor based on the Skylake architecture) and 16 GB RAM with 1 TB of HDD was used to carry out the experiments. The Ubuntu 18.04 and the Cuckoo Sandbox were installed along with additional Python scripts developed to extract all the information we need to perform the experiments. Windows 7 and Windows XP have been installed as the guest OS using the VirtualBox program where Cuckoo Sandbox submits files for analysis and retrieves different reports in JSON format. The virtual machine with Windows 7 has been mostly used for static analysis by IDA Pro and PEid to study the malware and retrieve the API call list. The purpose of this environment is to identify a group of API from malware and benign files and make a feature selection to be used for similarity to classify the given program as malicious or benign. Table 2 gives a summary of the dataset used in our experiments where the class column indicates whether the sample is benign or malware, the Quantity column gives the total number of each class, and the remaining two columns detail the distribution in terms of percentage and size (kilobyte).

**Table 2.** Dataset of malware and benign used in our experiment.

| Type or Class | Quantity | Percentage | Maximum Size (KB) |
|---|---|---|---|
| Benign | 17,214 | 21.91 | 122,154 |
| Trojan | 16,146 | 20.55 | 18,547 |
| Virus | 19,821 | 25.23 | 784 |
| Worm | 13,541 | 17.23 | 16,000 |
| Rootkit | 357 | 0.45 | 789 |
| Backdoor | 8541 | 10.87 | 2054 |
| Flooder | 1424 | 1.81 | 39,785 |
| Exploit | 1524 | 1.94 | 29,431 |

*4.2. API Call Frequency and Sequence of Occurrence*

From static and dynamic analysis, we capture the API function from malware and benign files and it is expressed as API call sequence. Let V denote the dataset that contains M as malware, PE as portable executable files, and BF as benign files, $V = \{M, PE, BF\}$. The frequency of every function is then computed in order to have the most significant API functions list. Let $f_{m_i} = \{m_1, m_2, m_3, \ldots, m_n\}$ be the frequencies of functions in the M and $f_{p_i} = \{p_1, p_2, p_3, \ldots, p_n\}$ be the frequencies in the PE, and $f_{b_i} = \{b_1, b_2, b_3, \ldots, b_n\}$ frequencies in the BF. Finally, let us denote the API function as A. Algorithm 1 describes the main steps used in our method.

**Definition 1.** *For each API function **A**, from **M**, **PE**, and **BF** datasets, there is a frequency f for each A denoted $m_i \in M$; $p_i \in PE \land b_i \in BF$ which measures the number of times that A occurs in V dataset. The computation of the frequencies is satisfied as follows:*

$$f_{m_i} = \frac{\sum m_i}{\sum M} \tag{1}$$

$$f_{p_i} = \frac{\sum p_i}{\sum PE} \tag{2}$$

$$f_{b_i} = \frac{\sum b_i}{\sum BF} \tag{3}$$

---

**Algorithm 1: API Call List Frequency and Sequence—Computation Process**

---

**Input:**      $V = \{M, PE, BF\}$
**Output:**
$f = \{m_i, p_i, b_i\}$      // Set of frequencies representing all A occurrence in V
    1.  **Run** V in Win 7 // Static Analysis by IDA Pro and PEid
    2.    **Create report** from IDA Pro and PEid // Preliminary database about API Function List
    3.    **Query** on the preliminary DB
    4.    **Verity if** the database **= Ø** // Empty database
    5.     **If yes, abort**
    6.     **Otherwise,**
    7.    **For** each A in the database,
    8.     **Compute** $f_{m_i}, f_{p_i}, f_{b_i}$
    9.     **Insert** ($f_{m_i}, f_{p_i}, f_{b_i}$) to API Sequence
    10.   **End for**
    11.  **Run** V in Windows XP // Cuckoo Sandbox for dynamic analysis
    12.    **Repeat 2** to **10**
    13.  **Duplicates removal:** API_Sequence, Max_Length_Repeat
    14.  **Database =** API Sequence
    15.  **Init:** SeqIndex ← 0;
    16.  **While true do:**
    17.   **If SeqIndex =** Length_Sequence then
    18.    break;
    19.  **Else**
    20.    **for i** ← 1 to Max_Length_Repeat do
    21.     **if** found_repeat = true, **then**
    22.      **break;**
    23.    **End**
    24.    Found_repeat ← remove (API_Sequence, SeqIndex, i);
    25.    SeqIndex +=i; // Update the API Sequence Database
    26.   **End**
    27.   **End**
    28.  **End**

---

### 4.3. Similarity Analysis and Classification

The cross method proposed in this research is based on static and dynamic analysis, where the executable malware is disassembled. Hereafter, the similarity analysis is performed for each dissembled executable (β), which represents the vector of functions m, n where (β′) is the variant malware of the original executable (β). Each function is represented as an array of vectors of functions. Therefore, we use (β) and (β′) to compute the similarity which gives us a measure as a reference for the similarity of coefficients with values between 0 and 1. Should the two vectors be similar, the value is 0 while should they not be similar, the value is 1. The obtained value is then compared to a given

threshold value (e.g., 0.5, 0.6, or 0.7). The threshold of similarity (δ) was chosen for the purpose of the empirical result [44–47].

In this research, different methods have been tested for similarity analysis:

- **Minkowski Distance**: *The Minkowski distance between two n-dimension vectors A and B is given by:*

$$D(A, B) = \left( i\sum_{i=1}^{n} |A_i - B_i|^p \right)^{1/p} \tag{4}$$

The Minkowski Distance can be considered as a generalization of both the Euclidean distance and the Manhattan distance. When $p = 1$, it corresponds to the Manhattan distance and when $p = 2$, it corresponds to the Euclidean distance.

- **Cosine Similarity:** Cosine similarity is a measure of similarity between two vectors based on the cosine of the angle between them. The vectors $A$ and $B$ are usually the term frequency vectors. The Cosine similarity between vectors $A$ and $B$ is given by:

$$similarity = \text{Cos}(\theta) = \frac{A.B}{||A||||B||} = \frac{\sum_{i=1}^{n} A_i x B_i}{\sqrt{\sum_{i=1}^{n} (A_i)^2} \sqrt{\sum_{i=1}^{n} (B_i)^2}} \tag{5}$$

The resulting similarity ranges from −1 meaning exactly opposite, to 1 meaning exactly the same, with 0 indicating orthogonality or decorrelation, while in-between values indicate intermediate similarity or dissimilarity.

- **Containment Broder:** Containment border defines the containment for comparing two documents. The function $f$ () computes sets of features from two documents, $h$ and $i$, such as fingerprints of "shingles". The *containment*($h$, $i$) of h within $i$ is defined as:

$$containment(h, i) = \frac{|f(h) \cap f(i)|}{|f(h)|} \tag{6}$$

- **Canberra Distance**: The Canberra distance is a weighted version of the Manhattan distance, introduced and refined in 1967 by Lance, Williams, and Adkins [48]. It is often used for data scattered around an origin, as it is biased for measures around the origin and very sensitive for values close to zero.

$$D = \frac{\sum_i |u_i - v_i|}{\sum_i |u_i + v_i|} \tag{7}$$

- **Longest Common Subsequence (LCS):** The *LCS* is used to find the longest subsequence common to all sequences in any two given strings $P$. To extract the common API call sequence pattern among malware, the longest common subsequences (LCSs) is used. The formula is shown in (8). In the formula, $P_i$ and $Q_i$ represent the *i*th character of sequences $P$ and $Q$, respectively. For example, the LCSs of TKMN and TMK are TK and TM:

$$LCS(P_i, Q_j) = \begin{cases} \varnothing & if\ i = 0\ or\ j = 0 \\ LCS(P_{i-1}, Q_{j-1}) & if\ p_i = q_j \\ longest(LCS(P_i, Q_{j-1}), LCS(P_{i-1}, Q_j)) & if\ p_i \neq q_j \end{cases} \tag{8}$$

### 4.4. Experiments Results

Table 3 shows the mean values of each of the methods to measure the similarity between malware. The mean values are then compared with the threshold value of 0.5; the threshold was chosen for empirical results. In this case, if a value of similarity exceeds the threshold, the malware under investigation is considered as a variant and therefore malicious. The Minkowisk distance shows a good result as it combines two methods: Manhattan and Euclidian distance, we have used Manhattan as it has been proven to have the best performance within the malware domain [34,35]. The measures of the similarity give a value range between 0 and 1, which means that the compared malware programs are similar if the value is 0, or dissimilar if the value is 1. Table 4 shows the result of a benign program without low or high values. However, in the case of Tables 5 and 6, we can see that there is a high similarity/low distance between malware variants. For more analysis, we have computed the mean of those distances. Furthermore, in the signature database, a value of each signature is computed and hence generates a report of similarity. Thereafter, the highest similarities are indexed, which indicated the most likely distinct malware variant or family. Finally, a decision is made as to whether the unknown executable is a variant or not.

Deep analysis on the similarity of two malware families (Banload and DyFuCa) are shown in Tables 5 and 6. We have selected Banload malware variants because its infection channel is based on the files downloaded from the internet by users when visiting malicious sites. For DyFuCa, it is essentially an intrusive adware or spyware that is hiddenly downloaded and installed on a computer. The highlighted cells in Tables 4–6 reveal that a malware variant is defined as having a distance less than or equal to 0.5. As shown, the entire cell in Table 5 can be detected as a variant of the original malware Win32.Bonload. For more details about the features, we have provided an explainable table in Appendix A.

**Table 3.** Similarity analysis mean matrix result [j × i].

| Method | Malware to Benign | Malware to Malware | Benign to Benign |
|---|---|---|---|
| Minkowski Distance | 168.32 | 101.14 | 198.04 |
| Cosine Similarity | 54.452 | 21.458 | 12.142 |
| Containment Broader | 86.652 | 45.256 | 10.254 |
| Canberra Distance | 10.845 | 9.154 | 7.501 |
| Longest Common Subsequence | 98.141 | 112.08 | 86.145 |

**Table 4.** Matrix of similarity for non-similar benign.

| Features | txt | ms | msgr | skype | ffing | xlxs | msppt | dialer |
|---|---|---|---|---|---|---|---|---|
| **txt** | 0.00 | 1.00 | 1.00 | 1.00 | 1.00 | 1.00 | 1.00 | 1.00 |
| **ms** | 1.00 | 0.00 | 1.00 | 1.00 | 1.00 | 1.00 | 1.00 | 1.00 |
| **msgr** | 1.00 | 1.00 | 0.00 | 1.00 | 1.00 | 1.00 | 1.00 | 1.00 |
| **skype** | 1.00 | 1.00 | 1.00 | 0.00 | 1.00 | 1.00 | 0.99 | 0.99 |
| **ffing** | 1.00 | 1.00 | 1.00 | 1.00 | 0.00 | 1.00 | 1.00 | 1.00 |
| **xlxs** | 1.00 | 0.98 | 1.00 | 1.00 | 1.00 | 0.00 | 1.00 | 1.00 |
| **msppt** | 1.00 | 1.00 | 1.00 | 1.00 | 1.00 | 1.00 | 0.00 | 1.00 |
| **dialer** | 1.00 | 1.00 | 1.00 | 0.98 | 1.00 | 1.00 | 1.00 | 0.00 |

**Table 5.** Similarity matrix of the Trojan: Win32/Banload.A family.

| Features | .texdt | .rdas | .rscf | .reloe | .kkng | .ipfi | .usere | .icon |
|---|---|---|---|---|---|---|---|---|
| **.texdt** | 0.00 | 0.14 | 0.24 | 0.10 | 0.10 | 0.10 | 0.10 | 0.10 |
| **.rdas** | 0.10 | 0.00 | 0.12 | 0.00 | 0.12 | 0.12 | 0.11 | 0.23 |
| **.rscf** | 0.21 | 0.23 | 0.00 | 0.25 | 0.22 | 0.21 | 1.00 | 0.28 |
| **.reloe** | 0.13 | 0.00 | 0.22 | 0.00 | 0.23 | 0.00 | 0.25 | 0.23 |
| **.kkng** | 0.13 | 0.16 | 0.21 | 0.26 | 0.00 | 0.00 | 0.00 | 0.21 |
| **.ipfi** | 0.13 | 0.16 | 0.22 | 0.26 | 0.00 | 0.00 | 0.00 | 0.16 |
| **.usere** | 0.14 | 0.16 | 0.28 | 0.26 | 0.00 | 0.00 | 0.00 | 0.16 |
| **.icon** | 0.14 | 0.15 | 0.21 | 0.26 | 0.19 | 023 | 0.21 | 0.00 |

**Table 6.** Matrix of similarity for the DyFuCa virus.

| Features | .jalr | .bgtz | .syscall | .fprem | .mrob | .ivtl | .usere | .srvbw |
|---|---|---|---|---|---|---|---|---|
| **.jalr** | 0.00 | 0.21 | 0.00 | 0.17 | 1.00 | 1.00 | 0.23 | 0.23 |
| **.bgtz** | 0.25 | 0.00 | 0.21 | 0.05 | 1.00 | 1.00 | 1.00 | 1.00 |
| **.syscall** | 0.21 | 0.15 | 0.00 | 1.00 | 1.00 | 1.00 | 1.00 | 0.09 |
| **.fprem** | 0.15 | 1.00 | 1.00 | 0.00 | 1.00 | 1.00 | 0.99 | 0.24 |
| **.mrob** | 0.21 | 1.00 | 1.00 | 1.00 | 0.00 | 1.00 | 1.00 | 0.07 |
| **.ivtl** | 1.00 | 0.98 | 0. | 1.00 | 1.00 | 0.00 | 1.00 | 1.00 |
| **.usere** | 1.00 | 0.28 | 0.11 | 0.18 | 1.00 | 1.00 | 0.00 | 0.05 |
| **.srvbw** | 1.00 | 0.18 | 0.11 | 0.18 | | | 0.05 | 0.00 |

From the above result, the malware variants within the same family present a high similarity while the experiments show that there is no similarity among the benign files, as is both expected and logical. Last but not least, it can be observed that there is a low similarity between the malware dataset and the benign dataset. For that reason, we can be assured that the proposed method is able to make a distinction between malware and benign datasets clearly. In conclusion, a similarity test can be applied to detect malware mutants and hence the use of such distance malware can be classified.

## 5. Conclusions

In this research, we showed that malware detection is possible using static and dynamic methods. In this paper, we proposed a method to classify malware based on an API call that reflects the behavior of a piece of malicious code. Furthermore, we introduced a method to construct features based on API sequence from the result of the static and dynamic analysis. Furthermore, the extracted features were correctly assigned by a computation of similarity-based machine learning algorithms and statistics methods to profile and classify the program files as benign or malware. Finally, the experimental results of the similarity analysis of malware based on API call and the frequency of appearance of executable files reveal that this technique is effective in the detection and classification malware even if obfuscation methods are applied on the malware. For future work, we plan to perform research on Windows 8 and 10 as these are the latest Operating Systems of Microsoft Windows. We also plan to target where to apply our research, such as Financial Institutions as they are more targeted by malware makers. Our main goal is to conduct research on malware and their variants that are specific to banks.

**Author Contributions:** Design, Concept, and First Version Article, B.N.; Experiment and Implementation K.H.K.; Data Analysis and Report Generation Y.J.K.; Code Review, Second Version Article and Compilation H.H.K.; Supervison, Project Manager T.Y.K.; Funding Acquisition, Investigation, Resources, H.J.L.

**Funding:** This research received no external funding.

**Acknowledgments:** This work was supported by Institute for Information and Communications Technology Promotion (IITP) grant funded by the Korea government (MSIT) (2018-0-00285, Development of HD video data and control signal Endecryption processing high confidence dual core SoC management system for Unmanned Vehicle) & (No. 2018-0-00245, Development of prevention technology against AI dysfunction induced by deception attack).

**Conflicts of Interest:** The authors declare no conflict of interest.

## Appendix A. Description of Feature Selection from Tables 4–6

| Opcode Name | Description | Opcode Name | Description |
|---|---|---|---|
| Txt | Set of Instruction for text file | .rdas | Library |
| Ms | Set of Instruction for Microsoft Word | .rscf | Library |
| Msgr | Set of Instruction on Messenger Application | .releo | Return Load Execution |
| Skype | Set of Instruction on Skype application | .kkng | Library |
| Ffing | Set of Instruction on this application | .ipfi | Library |
| xlxs | Set of Instruction on Excel | .usere | Driver |
| msppt | Set of Instruction on Power Point | icon | Driver |
| dialer | Set of Instruction on Dialer | jalr | Jump Instruction |
| .textdt | Library | bgtz | Branch Instruction |
| Syscall | Fast System Call | fprem | Partial Reminder |
| Mrob | Move Result Object | ivtl | Invoke Virtual |
| Usere | Driver | srvbw | Store Value Bytes |

## Appendix B. Source Code in Python to Computer Similarly between Malware

To identify malware samples which are related to one another, we have edited and used python code to compute the import hash called "imphash". The imphash can also be used to identify similar samples created by a certain threat group. The code is given as is. For our experiment, we have editedit to customize it according to our dataset

```python
import sys,os
import pefile
import hashlib
import xlsxwriter
  if __name__ == "__main__":
 #Identify specified folder with suspect files
  dir_path = sys.argv[1]
 #Create a list of files with full path
 file_list = []
 for folder, subfolder, files in os.walk (dir_path):
   for f in files:
     full_path = os.path.join (folder, f)
     file_list.append (full_path)
 #Open XLSX file for writing
 #CSV can be also use to output
 #file_name = "pefilename-csv" // Uncomment this line if you don't want
 #to use XLSX and edit the other lines
 file_name = "filename_output.xlsx"
 workbook = xlsxwriter.Workbook (file_name)
 bold = workbook.add_format ({'bold':True})
 worksheet = workbook.add_worksheet ()
 #Write column headings
 row = 0
 worksheet.write ('A1', 'SHA256', bold)
 worksheet.write ('B1', 'Imphash', bold)
 row += 1
```

```
  #Iterate through file_list to compute imphash and sha256 file hash
   for item in file_list:
      #Get sha256
     fh = open (item, "rb")
     data = fh.read ()
     fh.close ()
     sha256 = hashlib.sha256 (data).hexdigest ()
      #Get import table hash
     pe = pefile.PE (item)
     ihash = pe.get_imphash ()
      #Write hashes to doc
     worksheet.write (row, 0, sha256)
     worksheet.write (row, 1, ihash)
     row + = 1
 #Autofilter the xlsx file for easy viewing/sorting
  worksheet.autofilter (0, 0, row, 2)
  workbook.close ()
```

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
