# Peer review of "Cross-Method-Based Analysis and Classification of Malicious Behavior by API Calls Extraction"

_applsci, doi:10.3390/app9020239_

Round 1

Reviewer 1 Report

This was an interesting manuscript, with a number of interesting conclusions. However, there are a number of format and content suggestions:

Line 52 : Not shy what “Hwy” is and presume it should be “Why”?

Line 70: Grammar check – “In paper” should be “In this paper”

Line 76: Missing references.

Figure 1: This is a nice diagram but please include the difference between dashed arrow lines, thin and thick arrow lines.

Line 113 and Table 1: Please incorporate the percent of malware (i.e. 78% different malware; 12% benign) so the reader has an understanding of the imbalance in your sample.

Figure 2: The figures in the row ‘%’ are proportions not percentages and the figures are hard to read and comprehend. Suggest convert to percentages to 2 decimal places.

Line 145: Reference missing.

Line 149: Grammar – should be “as follows” not “as follow”.

Line180: Grammar – should be “the authors raise” not “the authors rise”.

Section 4: The authors mix up methods with results. These should be clear and separate sections in their manuscript.

Section 4.1: It is not clear why Windows 10 Virtualbox has not be tested?

Line 197: Please define your abbreviation of Windows (e.g. Windows XP (Win XP)).

Lines 213 to 214: Please provide a reference to your statement “because it provides a safe environment to execute malware without infecting the whole system”.

Line 221 (etc): You do not need to keep repeating your sample size.

Table 2: You have not referenced this in your text? Please include a column representing percentages and suggest replace ‘Qt’ with “N”.

Lines 273 to 274: Please provide a reference for the 1967 paper quoted.

Line 291: Spelling – “tom measure” should be “to measure”

Lines 292 to 293: The sentence surrounding thresholds is unclear, so please re-phrase so it is clearer that you used a threshold of 0.5. Please explain further why you used this threshold value?

Line 296: You introduce the two malware families but do not explain why you chose these and why they are important in your analyses?

Tables 4 to 6: Please provide definitions in an Appendix of the Features extensions so the reader has a better understanding of these tables.

5 Conclusions: You need to provide a section before your conclusion DISCUSSION to discuss the implications of your research.

Author Response

Dear Valued Reviewer,

Please find all the answers in the attached PDF file

We really appreciate your excellent work

Thank you and best regards,

Happy New Year 2019!

Reviewer 2 Report

The work focuses on the problem of unknown malware detection. Concretely, features extraction such API call sequences and code obfuscation features were used to study the behavior of the sample execution and malware such as file operations creation and deletion of process. This method is based on dynamic analysis to classify the malware variants with similar behavior context. The API sequence are best feature to use to computer similitude or similarity of two malwares because they represent the behavioral features of malware. Authors claim that their experimental results show that the malware detection accuracy is very useful to discover potential threats and can help the decision-maker to deploy appropriate countermeasures.

This is a very challenging topic and area and authors tried to prove their findings. However, there are several issues that need to be answered before the paper can be published:

In the introduction, authors fail to precisely include the basic intuition behind this work. They should prove the contribution in a more solid way.

The related work is too small and authors cannot state the differences among their work and the references.

Figure 1 is not discussed at all. For example, which are the ML algorithms? What is Benign?

Table 1 includes summary of the dataset. Only these features?

In subsection 3.2, authors state “In this research, to overcome the challenge of packed malware, we have used the dissembler tools such IDA Pro[40] and Pestudio [41].” In which way, these tools were used?

In table 2, are there different datasets employed?

Authors do not discuss about the way that they chose the specific features for Tables 4-6. Furthermore, they do not discuss the corresponding results.

Finally, authors could add a paragraph regarding their future work.

Author Response

Dear Estimated Reviewer,

Please find all the answers in the PDF attached file

We appreciate the excellent work you have done

Thank you and best regards,

Happy New Year 2019!

Reviewer 3 Report

line19, mention/state that it is a new or novel method

line22, say what the calls used as information provides insight into within that sentence

What aspect are you detecting in the malware behavior? A sentence or information within those sentences is needed to provide a hint at least into what new features is derived. 25% of the abstract is what is novel and the new results obtained.

line36, what 'kind of networks'? seems arbitrary and vague

l44, is dynamic domain name system DNS? the 'dynamic' part should differentiate it from the classic service, right?

l49, In the case of attacks or a successfully committed cybercrime...

l50, to investigate such malicious activites

l52, is it, 'Hwy'? or How?

l55, what is a signature-based detection system? a look up system based upon recorded malicious activities from previous attacks?

l68, 'The AP sequence is the best...', I would recommend that you change the gravity of the statement as this may not be the case all the time

l74 'consists'

l78 sequences

l78, expand on the design of the feature selection database

l80-82, some more insight into the quantitative or qualitative results should be given. By this point the reader still does not have any clue as to what impact is derived

line76, fix the references

l87, research

l87, ' and new research is still being performed on providing solutions'

l89, samples of activity against known malware ...

In general there needs to be more granularity in the descriptions and now as high level. A practitioner should be able to follow the details.

l97, although well known, it is customary to describe the background to cover it to some degree to make the paper 'self-contained', it may seem as redundant or repetitive but it is custom

l105-108, this is definitely not a sufficient description. Regardless of any references and overlaps, even if so, the reader must be provided a guided walk through the elements and connectivity of the system to understand its operation in detail. A publication is should offer a road map to be replicated

Fig2 caption, please provide a suitable description in the caption, the current state is an enigma

l179, what is this distance measure? even if section4 describes it, please provide the header description.

l315, please expand on the implications for practical use cases, this need not overlap with the conclusions section

for the appendix, please use the latex pseudo code packages for proper formatting

Author Response

Dear Estimated Reviewer,

Please find all the answers in the PFD attached file

We really appreciate the excellent work you have done

Thank you and best regards,

Happy New Year 2019!
